# Implicit regularization in Heavy-ball momentum accelerated stochastic gradient descent

**Avrajit Ghosh**[*]  **He Lyu** [*]  **Xitong Zhang**  **Rongrong Wang**
Department of Computational Mathematics, Science and Engineering (CMSE)
Michigan State University
Correspondance to: `ghoshavr@msu.edu`

## Abstract

It is well known that the finite step-size ($h$) in Gradient Descent (GD) implicitly regularizes solutions to flatter minima. A natural question to ask is *"Does the momentum parameter $\beta$ play a role in implicit regularization in Heavy-ball (H.B) momentum accelerated gradient descent (GD+M)?"* To answer this question, first, we show that the discrete H.B momentum update (GD+M) follows a continuous trajectory induced by a modified loss, which consists of an original loss and an implicit regularizer. Then, we show that this implicit regularizer for (GD+M) is stronger than that of (GD) by factor of ($\frac{1+\beta}{1-\beta}$), thus explaining why (GD+M) shows better generalization performance and higher test accuracy than (GD). Furthermore, we extend our analysis to the stochastic version of gradient descent with momentum (SGD+M) and characterize the continuous trajectory of the update of (SGD+M) in a pointwise sense. We explore the implicit regularization in (SGD+M) and (GD+M) through a series of experiments validating our theory.

## 1 Introduction

Deep neural networks (NN) have led to huge empirical successes in recent years across a wide variety of tasks, ranging from computer vision, natural language processing, autonomous driving to medical imaging, astronomy and physics (Bengio & LeCun, 2007; Hinton et al., 2006; Goodfellow et al., 2016). Most deep learning problems are in essence solving an over-parameterized, large-scale non-convex optimization problem. A mysterious phenomenon about NN that attracted much attention in the past few years is why NN generalizes so well. Indeed, even with extremely over-parametrized model, NNs rarely show a sign of over-fitting (Neyshabur, 2017). Thus far, studies along this line have successfully revealed many forms of implicit regularization that potentially lead to good generalization when gradient descent (GD) or stochastic gradient descent (SGD) algorithms are used for training, including norm penalty (Soudry et al., 2018), implicit gradient regularization (Barrett & Dherin, 2020), and implicit Hessian regularization (Orvieto et al., 2022a;b) through noise injection.

In contrast, the family of momentum accelerated gradient descent methods including Polyak's Heavy-ball momentum (Polyak, 1964), Nesterov's momentum (Sutskever et al., 2013), RMSProp (Tieleman et al., 2012), and Adam (Kingma & Ba, 2014), albeit being powerful alternatives to SGD with faster convergence rates, are far from well-understood in the aspect of implicit regularization. In this paper, we analyze the implicit gradient regularization in the Heavy-ball momentum accelerated SGD (SGD+M) algorithm with the goal of gaining more theoretical insights on how momentum affects the generalization performance of SGD, and why it tends to introduce a variance reduction effect whose strength increases with the momentum parameter.

## 2 Related literature

It has been well studied that gradient based optimization implicitly biases solutions towards models of lower complexity which encourages better generalization. For example, in an over-parameterized quadratic model, gradient descent with a near-zero initialization implicitly biases solutions towards

---

[*]denotes equal contribution

having a small nuclear norm (Arora et al., 2019; Gunasekar et al., 2017; Razin & Cohen, 2020), in a least-squares regression problem, gradient descent solutions with 0 initial guess are biased towards having a minimum $\ell_2$ norm (Soudry et al., 2018; Neyshabur et al., 2014; Ji & Telgarsky, 2019; Poggio et al., 2020). Similarly, in a linear classification problem with separable data, the solution of gradient descent is biased towards the max-margin (i.e., the minimum $\ell_2$ norm) solution (Soudry et al., 2018). However in (Vardi & Shamir, 2021), the authors showed that these norm-based regularization results proved on simple settings might not extend to non-linear neural networks.

The first general implicit regularization for GD discovered for all non-linear models (including neural networks) is the Implicit Gradient Regularization (IGR) (Barrett & Dherin, 2020). It is shown that the learning rate in gradient descent (GD) penalizes the second moment of the loss gradients, hence encouraging discovery of flatter optima. Flatter optima usually give higher test-accuracy and are more robust to parameter perturbations (Barrett & Dherin, 2020).

Implicit Gradient Regularization was also discovered for Stochastic Gradient Descent (SGD) (Smith et al., 2021) Li et al. (2019) , as one (but perhaps not the only one) reason for its good generalization. SGD is believed to also benefit from its stochasticity, which might act as a type of noise injection to enhance the performance. Indeed, it is shown in (Wu et al., 2020) that, by injecting noise to the gradients, full-batch gradient descent will be able to match the performance of SGD with small batch sizes. Besides injecting noise to the gradients, many other ways of noise injections have been discovered to have an implicit regularization effect on the model parameters, including noise injection to the model space (Orvieto et al., 2022b) and those to the network activations (Camuto et al., 2020). However, how these different types of regularization cooperatively affect generalization is still quite unclear.

The effect of generalization in momentum accelerated gradient descent has been studied much less. Li et al. (2019) analyzed the trajectory of SGD+M and found that it can be weakly approximated by solutions of certain Ito stochastic differential equations, which hinted the existence of IGR in (SGD+M). However, both the explicit formula of IGR and its relation to generalization remain unknown. Recently, in (Wang et al., 2021), the authors analyzed the implicit regularization in momentum (GD+M) based on a linear classification problem with separable data and show that (GD+M) converges to the $L_2$ max-margin solution. Although this is one of the first proposed forms of implicit regularization for momentum based methods, it fails to provide an insight on the implicit regularization for momentum in non-linear neural networks.

Recently, (Jelassi & Li, 2022) has shown that the (GD+M) increases the generalization capacity of networks in some special settings (i.e., a simple binary classification problem with a two layer network and part of the input features are much weaker than the rest), but it is unclear to which extent the insight obtained from this special setting can be extended to practical NN models.

To the best of our knowledge, no prior work has derived an implicit regularization for (SGD+M) for general non-linear neural networks.

## 3    IMPLICIT GRADIENT REGULARIZATION FOR GRADIENT DESCENT AND ITS RELATION TO GENERALIZATION

We briefly review the IGR defined for GD (Barrett & Dherin, 2020) which our analysis will be based on. Let $E(\mathbf{x})$ be the loss function defined over the parameters space $\mathbf{x} \in \mathbb{R}^p$ of the neural network. Gradient descent iterates take a discrete step ($h$) opposite to the gradient of the loss at the current iterate

$$\mathbf{x}^{k+1} = \mathbf{x}^k - h\nabla E(\mathbf{x}^k). \tag{1}$$

With an infinitesimal step-size ($h \to 0$), the trajectory of GD converges to that of the first order ODE

$$\mathbf{x}'(t) = -\nabla E(\mathbf{x}(t)) \tag{2}$$

known as the gradient flow. But for a finite (albeit small) step size $h$, the updates of GD steps off the path of gradient flow and follow more closely the path of a modified flow:

$$\mathbf{x}'(t) = -\nabla \hat{E}(\mathbf{x}(t)), \qquad \text{where } \hat{E}(\mathbf{x}) = E(\mathbf{x}) + \frac{h}{4}\|\nabla E(\mathbf{x})\|^2. \tag{3}$$

It is shown (Barrett & Dherin, 2020) via the so-called classical backward analysis that when GD and the two gradient flows 2 and 3 all set off from the same point $\mathbf{x}^k$, the next gradient update $\mathbf{x}^{k+1}$ is $O(h^2)$ close to the original gradient flow (2) evaluated at the next time point $t_{k+1} = h + t_k$, but is $O(h^3)$ close to the modified flow (3) evaluated at $t_{k+1}$. So, locally, the modified flow tracks the gradient descent trajectory more closely than the original flow. To discuss the global behaviour of GD, we need the following definition of closeness between two trajectories.

**Definition 3.1** ($O(h^\alpha)$-closeness in the strong sense). Fix some $T > 0$, we call the trajectory of the discrete GD-update $\mathbf{x}^k$ and a continuous flow $\tilde{\mathbf{x}}(t_k)$ to be $O(h^\alpha)$-close in the strong sense [1] if:

$$\max_{k \in \mathcal{K}} \|\mathbf{x}^k - \tilde{\mathbf{x}}(t_k)\|_2 \leq ch^\alpha, \quad \mathcal{K} = \left\{ 1, \dots \left\lfloor \frac{T}{h} \right\rfloor \right\},$$

where $t_k = hk$, and $c$ is some constant independent of $h$ and $k$

Definition 3.1 quantifies the global closeness of a discrete trajectory and a continuous one. By accumulating the local error, one can show that setting off from the same location $\mathbf{x}^0$, the original gradient flow equation 2 is $O(h)$-close to the GD trajectory while that of the modified flow 3 is $O(h^2)$-close. Based on this observation, the authors defined the $O(h)$ term $\frac{h}{4}\|\nabla E(\mathbf{x})\|^2$ in the modified flow as the IGR term and concluded by stating that it guides the solutions to flatter minima in a highly non-convex landscape.

To justify why minimizing $\|\nabla E(\mathbf{x})\|^2$ is a good idea and why it encourages flat minimizers (which seems to be missing from the original paper), we borrow an argument from (Foret et al., 2020) that was originally developed for a different purpose.

Due to the PAC-Bayes analysis (Neyshabur, 2017), a simplified generalization bound for NN derived under some technical conditions can be stated as (Foret et al., 2020)

$$\mathcal{L}_\mathcal{D}(\mathbf{x}) \leq \max_{\|\epsilon\| \leq \rho} \mathcal{L}_\mathcal{S}(\mathbf{x} + \epsilon) + \hat{h}\left( \frac{\|\mathbf{x}\|^2}{\rho} \right), \text{ for any } \rho > 0,$$

where $\mathcal{L}_\mathcal{D}$ is the population loss (i.e., the generalization error), $\mathcal{L}_\mathcal{S}$ is the empirical/training loss, and $\hat{h} : \mathbb{R}_+ \to \mathbb{R}_+$ is some strictly increasing function. One can try to minimize this upper bound in order to minimize the generalization error $\mathcal{L}_\mathcal{D}$. The term $\hat{h}\left( \frac{\|\mathbf{x}\|^2}{\rho} \right)$ in the upper bound can be controlled by activating a weight decay penalty during training, and the first term in the bound is usually written into

$$\max_{\|\epsilon\| \leq \rho} \mathcal{L}_\mathcal{S}(\mathbf{x} + \epsilon) = \underbrace{\max_{\|\epsilon\| \leq \rho} (\mathcal{L}_\mathcal{S}(\mathbf{x} + \epsilon) - \mathcal{L}_\mathcal{S}(\mathbf{x}))}_{\text{sharpness}} + \underbrace{\mathcal{L}_\mathcal{S}(\mathbf{x})}_{\text{training loss}}$$

which consists of the training loss and an extra term called sharpness, minimizing which will help with generalization. Since directly minimizing the sharpness is difficult, one can then use the following first-order Taylor approximation

$$\max_{\|\epsilon\| \leq \rho} (\mathcal{L}_\mathcal{S}(\mathbf{x} + \epsilon) - \mathcal{L}_\mathcal{S}(\mathbf{x})) \approx \max_{\|\epsilon\| \leq \rho} \epsilon^T \nabla \mathcal{L}_\mathcal{S}(\mathbf{x}) = \rho \|\nabla \mathcal{L}_\mathcal{S}(\mathbf{x})\|.$$

Using our notation, $\nabla \mathcal{L}_\mathcal{S}(\mathbf{x})$ is $\nabla E(\mathbf{x})$, so the sharpness is approximately proportional to $\|\nabla E\|$ which is the square root of the IGR term.

## 4 IMPLICIT GRADIENT REGULARIZATION FOR HEAVY BALL ACCELERATED GRADIENT DESCENT

The main mathematical challenge in studying the IGR in momentum updates is that we now need to perform global error analysis instead of the local backward analysis, as the momentum updates utilizes the entire update history.

---

[1] Weak-sense approximation as studied in Li et al. (2019) only requires the distributions of the sample processes $\mathbf{x}$ and $\tilde{\mathbf{x}}$ to be close, whereas our strong-sense approximation requires each instance of $\mathbf{x}$ and $\tilde{\mathbf{x}}$ to be close. The strong-sense IGR found using the latter is valid for trainings with any fixed random-batch sequence and any fixed initialization, while the former only characterizes the mean trajectory taking expectation over many different trainings (each with a random batch-sequence and initialization).

The IGR for Heavy-ball momentum was previously analyzed in (Kovachki & Stuart, 2021) through studying its relationship with the damped second order Hamiltonian dynamic

$$m\mathbf{x}'(t) + \gamma\mathbf{x}''(t) + \nabla E(\mathbf{x}(t)) = 0$$

which has been well-known as the underlying ODE for the momentum updates. However, only $O(h)$ closeness is proven between the momentum updates and this ODE trajectory under general step size assumptions [2], which is not enough since the implicit regularization term $\frac{h}{4}\|\nabla E\|^2$ itself is of order $O(h)$. In addition, this approach is difficult to be applied to the stochastic setting.

In this paper, we circumvent the use of the second order ODE (as it only gives $O(h)$ closeness) and directly obtain a continuous path that is $O(h^2)$-close to the momentum update for both GD and SGD. This is achieved by linking the momentum updates with a first order piecewise ODE, proving that the ODE has a piece-wise differentiable trajectory that is $O(h^2)$-close to the momentum updates, and then using its trajectory to study the IGR. The detailed argument can be found in the appendix. Here we provide the final mathematical formula for the implicit regularization of (Heavy-Ball) momentum based gradient descent method (IGR-M).

**Theorem 4.1.** *(IGR-M): Let the loss for the full-batch gradient $E$ be smooth and 4-times differentiable, then the (GD+M) updates*

$$\mathbf{x}^{k+1} = \mathbf{x}^k - h\nabla E(\mathbf{x}^k) + \beta(\mathbf{x}^k - \mathbf{x}^{k-1}) \quad \forall k = 1, 2, ..., n$$

*are $O(h^2)$ close to the flow of the continuous trajectory of the piecewise first-order ODE*

$$\widetilde{\mathbf{x}}'(t) = -\frac{1-\beta^{k+1}}{1-\beta}\nabla E(\widetilde{\mathbf{x}}(t)) - \frac{h\gamma(1+\beta)}{2(1-\beta)^3}\nabla^2 E(\widetilde{\mathbf{x}}(t))\nabla E(\widetilde{\mathbf{x}}(t)), t \in [t_k, t_{k+1}] \tag{4}$$

*where $t_k = kh$ and*

$$\gamma = (1 - \beta^{2k+2}) - 4(k+1)\beta^{k+1}\frac{(1-\beta)}{(1+\beta)}.$$

*Since $\beta^k$ quickly decays to 0 as $k$ grows, for a sufficiently large iteration $k$, equation 4 reduces to:*

$$\widetilde{\mathbf{x}}'(t) = -\frac{1}{1-\beta}\nabla\hat{E}(\widetilde{\mathbf{x}}(t)), \quad t \in [0, T] \tag{5}$$

*driven by the modified loss $\hat{E}(\widetilde{\mathbf{x}}(t)) := E(\widetilde{\mathbf{x}}(t)) + \frac{(1+\beta)h}{4(1-\beta)^2}\|\nabla E(\widetilde{\mathbf{x}}(t))\|_2^2$. More specifically, for a fixed time $T$, there exists a constant $C$, such that for any learning rate $0 < h < T$, we have*

$$\|\widetilde{\mathbf{x}}(t_k) - \mathbf{x}^k\|_2 \leq Ch^2, \ t_k = kh, \quad k = 1, 2, ..., \lfloor\frac{T}{h}\rfloor. \tag{6}$$

Comparing the continuous trajectory traced with 5 and the one without momentum 3, we immediately have a few important observations:

**Remark 4.1.** Ignoring the $O(h)$ terms in 5 and 3, we see that the momentum trajectory is driven by a force that is $\frac{1}{1-\beta}$ times as large as that for GD. Therefore, using the same learning rate, (GD+M) is expected to converge $\frac{1}{1-\beta}$ times as fast as GD. Alternatively, (GD+M) with a learning rate $h$ has roughly the same convergence rate as GD with a learning rate $\frac{h}{1-\beta}$. From now on, we call $\frac{h}{1-\beta}$ the effective learning rate of (GD+M).

**Remark 4.2.** In terms of the IGR, we can see that adding the momentum amplifies the strength of the IGR (the coefficient in front of the IGR term increased from the $\frac{h}{4}$ in (GD) to the $\frac{h}{4}\frac{1+\beta}{(1-\beta)^2}$ in (GD+M). Even when letting the effective learning rates of the two methods equal (i.e., $\frac{h_{GD+M}}{1-\beta} = h_{GD}$), the implicit regularization in (GD+M) is still $\frac{1+\beta}{1-\beta}$ times stronger.

**Remark 4.3.** The IGR for (GD+M) reduces to the IGR for GD when $\beta = 0$.

With an additional momentum parameter $\beta$, the strength of the implicit regularizer increases by a factor of $\frac{1+\beta}{1-\beta}$. Hence for increasing values of momentum parameter $\beta$, the strength of the regularization increases, thus preferring more flatter trajectories through its descent.

---

[2]An $O(h^2)$ closeness is proven under very stringent conditions on the learning rate, which excludes the interesting regime where momentum has an advantage over plain GD in terms of the convergence rate and stability

## 4.1 IGR-M in a 2D Linear Model

We first show the impact of IGR-M in a very simple setting that minimizes a loss function with two learnable parameters, i.e., $(\hat{w}_1, \hat{w}_2) = \arg\min_{w_1, w_2} E(w_1, w_2)$ where $E(w_1, w_2) = \frac{1}{2}(y - w_1 w_2 x)^2$. Here $x, y, w_1, w_2$ are all scalars and mimics a simple scalar linear two-layer network. For a given scalar $(x, y)$, the global minima of $E(w_1, w_2)$ are all the points on the curve $w_1 w_2 = \frac{y}{x}$. The continuous gradient flow of the parameters are $w_1'(t) = -\frac{\partial E}{\partial w_1}$ and $w_2'(t) = -\frac{\partial E}{\partial w_2}$.

The IGR for this problem is $\frac{h_{GD}}{4}\|\nabla E\|_2^2 = \frac{h_{GD}}{4}(w_1^2 + w_2^2)E$, which will regularize the trajectory to find minima with a smaller value $(w_1^2 + w_2^2)$ (towards the cross) among all the global minima lying on $w_1 w_2 = \frac{y}{x}$. We intentionally chose the same experiment as in (Barrett & Dherin, 2020) to compare the effect of implicit regularization for (GD) and (GD+M).

For a fair comparison between (GD) and (GD+M), we set the effective learning rates to be the same, i.e, $h_{GD} = \frac{h_{GD+M}}{1-\beta}$ as in Remark 4.1. With the same initialization $(w_1^0, w_2^0) = (2.8, 3.4)$ we explore and track the path of four trajectories with $(h_{GD+M}, \beta)$ being $(5 \times 10^{-3}, 0.5)$, $(2 \times 10^{-3}, 0.8)$, $(10^{-3}, 0.9)$ and $(10^{-2}, 0.0)$. In all the four cases, the effective learning rates are the same, i.e, $\frac{h_{GD+M}}{(1-\beta)} = h_{GD} = 10^{-2}$. We make the following observations: a) For all the four trajectories, the converged weight parameters $w^* = (w_1^*, w_2^*)$ lie on the global minima curve. b) With increasing value of $\beta$, the converged solutions have decreasing value of $\ell_2$ norm (or increasing strength of im-

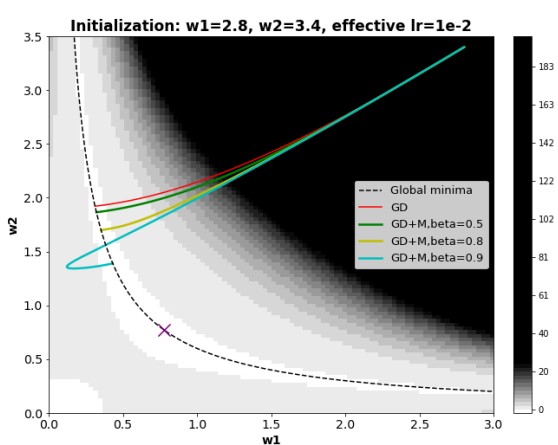

Figure 1: Implicit regularization for (GD+M) is stronger than that of (GD) for the same effective learning rate $\frac{h}{(1-\beta)}$. As $\beta$ increases the optima seems to find solution with a lower norm. This confirms Remark 4.2 that the strength of implicit regularization increases with $\beta$. The background color denotes the magnitude of the norm of the gradient, i.e. , $\|\nabla E\|_2^2$

plicit regularization), i.e, $\|w_{(10^{-3}, 0.9)}^*\|_2 < \|w_{(2 \times 10^{-3}, 0.8)}^*\|_2 < \|w_{(5 \times 10^{-3}, 0.5)}^*\|_2 < \|w_{(10^{-2}, 0.0)}^*\|_2$. This observation supports Remark 4.2, that the strength of implicit regularization increases with $\beta$, even with the effective learning rate.

## 5 Implicit Regularization in SGD with momentum

In SGD, the full-batch gradient is replaced by it's sampled unbiased estimator. Assume that the loss function $E(\mathbf{x})$ has the following form

$$E(\mathbf{x}) = \frac{1}{M}\sum_{j=1}^{M} E_{(j)}(\mathbf{x}) \tag{7}$$

where $E_{(j)}$ is the $j^{th}$ mini-batch loss. In the $k^{th}$ iteration, we randomly pick a mini-batch, whose loss is denoted by $E_k$, and update the parameters accordingly. The heavy-ball accelerated SGD iterates as

$$\begin{cases} \mathbf{x}^{k+1} = \mathbf{x}^k - h\nabla E_k(\mathbf{x}^k) + \beta(\mathbf{x}^k - \mathbf{x}^{k-1}) & k = 1, 2, ..., n \\ \mathbf{x}^1 = \mathbf{x}^0 - h\nabla E_0(\mathbf{x}^0) \\ \mathbf{x}^0 = \mathbf{x}^{-1} = \mathbf{0} \end{cases} \tag{8}$$

For each iteration $k$, the update is driven by the current mini-batch loss $E_k$. Its continuous approximation is

$$\mathbf{x}'(t) = -\nabla E_k(\mathbf{x}(t)) + \frac{\beta}{h}(\mathbf{x}(t_k) - \mathbf{x}(t_{k-1})) \quad \text{for } t_k < t < t_{k+1} \tag{9}$$

during the time $[t_k, t_{k+1}]$. As a result, the trajectory of $\mathbf{x}$ is continuous but piece-wise differentiable as it is easy to see that the left and right-side derivatives are not equal at a transit point $t_k$ from one batch to another, $\mathbf{x}'(t_k^+) \neq \mathbf{x}'(t_k^-)$. Therefore, we expect the implicit regularization term to also have discontinuous derivatives on different intervals. Below we present the mathematical formula for IGR-M in the stochastic setting.

**Theorem 5.1.** *[IGR-M stochastic version (IGRM$_s$)] Let the loss for each mini-batch $E_k$ be 4-times differentiable, then the Heavy Ball momentum updates 8 are $O(h^2)$ close to the trajectory of the gradient flow with the modified loss,*

$$\widetilde{\mathbf{x}}'(t) = -\nabla \hat{E}_k(\widetilde{\mathbf{x}}(t)) \quad \text{for } t_k \leq t < t_{k+1}$$

$$\text{where,} \quad \hat{E}_k(\widetilde{\mathbf{x}}) = \underbrace{G_k(\widetilde{\mathbf{x}})}_{force} + \underbrace{\frac{h}{4}(\|\nabla G_k(\widetilde{\mathbf{x}})\|_2^2 + 2\sum_{r=0}^{k-1} \beta^{k-r} \|\nabla G_r(\widetilde{\mathbf{x}})\|_2^2)}_{IGRM_s} \tag{10}$$

*with $G_k(\widetilde{\mathbf{x}}(t)) = \sum_{r=0}^{k} \beta^{k-r} E_r(\widetilde{\mathbf{x}}(t))$. Specifically, there exists a constant $C$ such that*

$$\|\widetilde{\mathbf{x}}(t_k) - \mathbf{x}^k\|_2 \leq Ch^2, \quad k = 1, 2, ..., n.$$

The theorem tells us that the IGR for momentum is in the form of $\ell_2$ norms of $\nabla G_k$ which is a weighted sum of the historical gradients $\nabla E_k$, $k = 0, ..., n$, by powers of $\beta$ and evaluated at the current location $\tilde{\mathbf{x}}(t)$. In addition, the stochastic IGR-M has different expressions on different intervals, caused by the abrupt changes between adjacent batches. Some further remarks about the (SGD+M) algorithm are listed below.

**Remark 5.1.** The flow of the continuous trajectory is governed by the driving-force $-\nabla G_k(\mathbf{x}(t))$ and the negative gradient of an implicit regularizer $IGRM_s(\widetilde{\mathbf{x}}) = \frac{h}{4}(\|\nabla G_k(\widetilde{\mathbf{x}})\|_2^2 + 2\sum_{r=0}^{k-1} \beta^{k-r} \|\nabla G_r(\widetilde{\mathbf{x}})\|_2^2$ which depends on both the learning rate $h$ and momentum $\beta$.

**Remark 5.2.** When $\beta = 0$, 10 reduces to

$$\widetilde{\mathbf{x}}'(t) = -\nabla \left( \underbrace{E_k(\widetilde{\mathbf{x}}(t)}_{force} + \underbrace{\frac{h}{4}\|\nabla E_k(\widetilde{\mathbf{x}}(t))\|_2^2}_{IGR} \right), \quad t \in [t_k, t_{k+1}], \tag{11}$$

which is the modified loss for SGD.

**Remark 5.3.** Taking expectation over the random selections of batches, we get $\mathbb{E}(IGRM_s)(\mathbf{x}) = \frac{h(1+\beta)}{4(1-\beta)^3}\|\nabla E\|^2 + \frac{h}{4(1-\beta)^2}F$, where $F := \mathbb{E}\|E_n - E\|^2$ (appendix Th 3.1). In comparison, the IGR term in SGD after taking expectation is $\frac{h}{4}(\|\nabla E\|^2 + F)$ (Smith et al., 2021), which is much weaker. Even with the adjusted learning rate, the IGR in (SGD+M) is still about $1/(1-\beta)$ times stronger than SGD.

**Remark 5.4** (Variance reduction)**.** We notice that momentum has a variance reduction effect. Explicitly, suppose the effective learning rate (Remark 4.1) is used so that the force terms in (SGD) and (SGD+M) have the same expectation, and then we can compare their variance. Let the covariance matrix of $\nabla E_k$ at a fixed point $\mathbf{x}$ be $C := \mathbb{E}(\nabla E_k(\mathbf{x}) - \nabla E(\mathbf{x}))(\nabla E_k(\mathbf{x}) - \nabla E(\mathbf{x}))^T$. Here $\nabla E(\mathbf{x})$ denotes the full-batch gradient. Then the covariance matrix of the force $-E_k$ driving (SGD) is exactly $C$, while that of the force $-G_k$ driving (SGD+M) is only $\frac{(1-\beta)}{1+\beta}C$ (appendix Th 4.1), which is about $\frac{(1-\beta)}{(1+\beta)}$ times smaller.

**Remark 5.5.** It is observed and confirmed by many literature that a larger variance of SGD iterations (caused by either a small batch size or a larger learning rate (Smith & Le, 2017; Li et al., 2017)) increases the generalization power. Larger variance in mini-batch gradients increases the escape efficiency of SGD from bad local minimas (Ibayashi & Imaizumi, 2022) [See Appendix section-7 for detailed discussion], hence increasing generalization power. Therefore, the variance reduction effect of (SGD+M) hurts generalization. On the other hand, the fact that (SGD+M) has a stronger IGR (Remark 5.3) and that (SGD+M) is more stable than (SGD) to the use of large effective learning rates (see e.g., (Cohen et al., 2021)) tend to help with its generalization. This explains why in practice we usually observe that (SGD+M) is only slightly better than (SGD).

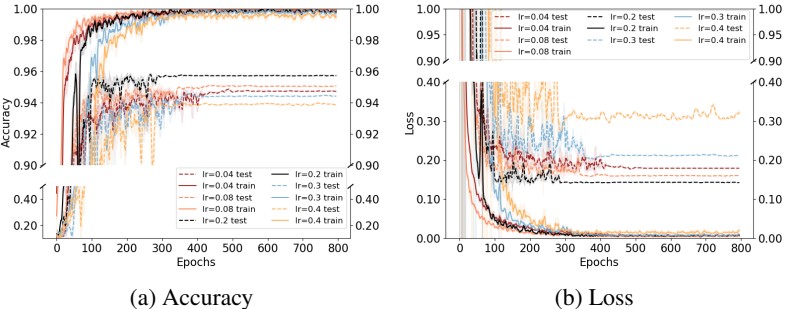

(a) Accuracy          (b) Loss

Figure 2: (GD): Classification results of ResNet-18 on MNIST dataset performed with full-batch gradient descent. Figure shows the effect of implicit regularization due to the finite learning rate $h$. Test accuracy improves with higher learning rate till $h = 0.2$. For $h = 0.2$, the best test-accuracy is achieved at $95.72\%$

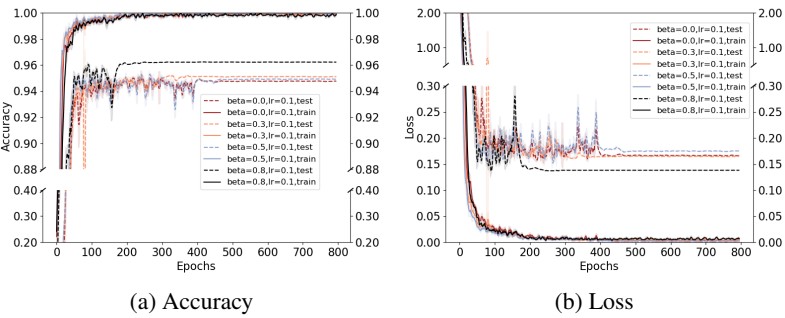

(a) Accuracy          (b) Loss

Figure 3: (GD+M): Classification results of ResNet-18 on MNIST dataset trained with various values of momentum parameter $\beta$ for full-batch gradient descent. The best test-accuracy is reported to be $96.22\%$

## 6 NUMERICAL EXPERIMENTS

Our first experiment is to compare the full-batch (GD) with (GD+M). For a linear least-squares problem with a Hessian matrix bounded by $L$ in the spectral norm, it is well-known that (e.g., (Cohen et al., 2021)) (GD+M) is stable as long as $h \leq \frac{2+2\beta}{L}$, and GD is stable as long as $h \leq \frac{2}{L}$. This means, the maximum achievable effective learning rate by (GD) is $\frac{2}{L}$, while that by (GD+M) can be as large as $\frac{2(1+\beta)}{L(1-\beta)}$. Since larger effective learning rates means a stronger IGR, (GD+M) clearly benefits from its large stability region. To confirm this, ResNet-18 is used to classify a uniformly sub-sampled MNIST dataset with 1000 training images. All external regularization schemes except learning rate decay and batch normalization have been turned off. We perform the following experiments : **1**) Full-batch gradient descent with $\beta = 0$ (i.e., GD) for various learning rate $h$ and the best test accuracy is noted (in Figure 2) to be $95.72\%$. **2**) Full-batch gradient descent with momentum (GD+M) performed for various $\beta$ with a fixed step-size $h = 0.1$ and the best test-accuracy is noted (in Figure 3) to be $96.22\%$. Our observation is that the best performance of GD (across all learning rates) is worse than the best performance of (GD+M) (across all $\beta$'s). This observation failed to be explained by the known theory of edge of stability[3] but can be well-explained by our implicit regularization theory for (GD+M) as adding momentum increases the strength of the IGR.

To study the effect of implicit regularization (SGD+M), a series of experiments have been performed on an image classification task. Four well-known and popular network architectures

---

[3]"edge of stability" (EOS) Cohen et al. (2021) is a phenomenon that shows during network training by the full batch gradient descent, the sharpness $||\nabla^2 E||_2$ tends to progressively increase until it reaches the threshold $\frac{2}{h}$ and then hovers around it. For GD+M, the sharpness will hover around a large value $\frac{2(1+\beta)}{h}$. Since larger sharpness usually means worse generalization, the EOS theory then predicts that adding momentum hurts the generalization.

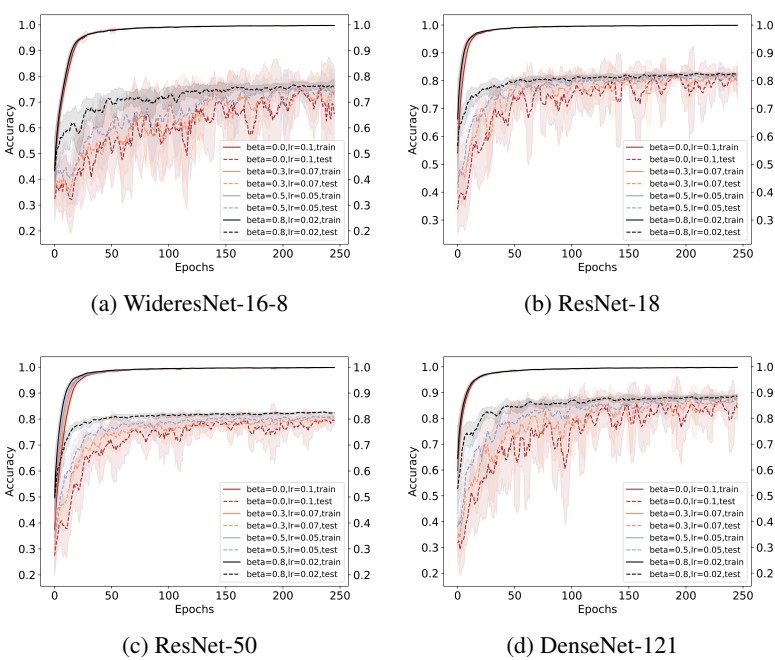

Figure 4: Classification results for CIFAR-10 dataset with various network architectures with combinations of $(h, \beta)$ chosen such that the effective learning rate $\frac{h}{(1-\beta)}$ remains same. In all of the experiments, external regularization like weight-decay, l.r scheduler, dropout,label-smoothing are kept off (except Batch-normalization). The results have been averaged over 3 random seeds having different initializations. (SGD+M) has a) higher test accuracy for increasing $\beta$ than (SGD) confirming Remark 5.4 b) Less variance for test accuracy confirming Remark 5.3.

.

namely DenseNet (Iandola et al., 2014), ResNet-18, ResNet-50 (He et al., 2016) and WideResNet (Zagoruyko & Komodakis, 2016) are trained to classify images from the CIFAR-10 and CIFAR-100 datasets. We are interested to know how well training these networks with (SGD) and (SGD+M) respectively can generalize well onto the test dataset. To solely observe the effects of the momentum parameter $\beta$ and learning rate $h$ in generalization, we turn off all the external regularization like dropout, weight-decay and label-smoothing. We fix the batch-size to 640 in all our experiments.

| | CIFAR-10 | | | | CIFAR-100 | | | |
|---|---|---|---|---|---|---|---|---|
| $\beta$ /$h$ | DN-121 | RN-18 | RN-50 | WRN-16-8 | DN-121 | RN-18 | RN-50 | WRN-16-8 |
| 0.0/0.10 | 84.0±5.0 | 79.7±5.6 | 79.3±2.5 | 65.3±18.1 | 60.4±4.9 | 53.1±0.6 | 47.4±2.1 | 38.6±3.8 |
| 0.3/0.07 | 85.1±5.4 | 78.7±9.3 | 80.0±1.5 | 72.5±7.4 | 60.0±8.6 | 52.7±1.0 | 48.9±2.2 | 37.0±6.1 |
| 0.5/0.05 | 87.6±1.2 | 81.5±0.9 | 80.7±0.7 | 71.8±9.8 | 63.2±2.4 | 53.3±1.1 | **50.3±1.0** | 39.4±3.9 |
| 0.8/0.02 | **88.6±0.7** | **82.4±0.4** | **82.4±0.7** | **75.4±2.8** | **64.7±0.8** | **54.3±0.6** | 49.6±0.7 | **40.6±1.2** |

Table 1: Testing accuracy of CIFAR-10 and CIFAR-100 with different momentum $\beta$ and learning rates $h$, but the same effective learning rate $\frac{h}{(1-\beta)} = 0.1$. The best performance of different models is highlighted. The mean and the standard deviation of test accuracy is calculated over the last 5 epochs and three random seed initializations. DN-121 is the Densenet-121, RN-18 and RN-50 denote the Resnet-18 and Resnet-50, WRN-16-8 represents the WideResnet with depth 16 and width-factor 8.

In the first experiment, we showed (GD+M) has a larger stability region than (GD) and hence allows for the use of a larger effective learning rate. The same conclusion holds for (SGD+M) and (SGD). However, here we want to show that even in the region where both algorithms are stable, (SGD+M) is still not just a scaled version of (SGD). For this purpose, we pick a small learning rate to ensure stability of both algorithms, and keep the effective learning rate $\frac{h}{(1-\beta)}$ for (SGD+M) to be the same as the learning rate for (SGD) (both equal 0.1). We observe from Table 1 (also Figure 4), that the

maximum test accuracy is almost always achieved at the highest value for $\beta$. This observation is consistent with Remark-5.3 where we showed that the implicit regularization in (SGD+M) is indeed stronger than (SGD), even after the learning rate adjustment.

The standard deviation of test accuracy in Table 1 is calculated over the last 5 epochs and three random seed initialization. Lower standard deviation indicates a smoother test accuracy curve meaning less variation of test accuracy within an epoch interval. We observe that the lowest standard deviation is achieved at the highest value of $\beta$. Hence the observation that variance reduction effect is more prominent with higher $\beta$ is consistent with Remark 5.4.

## 7 COMBINED EFFECTS OF IGR AND NOISE INJECTION

Despite its close relation to sharpness (Section 3), the IGR term $\|\nabla E\|^2$ gets very weak and irrelevant as $\mathbf{x}^k$ approaches a local minimizer, since $\nabla E \to 0$. However, we find that this would not be the case if there was noise injection, which can help the IGR term retain its power even near local minima. More specifically, as studied in previous literature (Orvieto et al., 2022b; Camuto et al., 2020), the algorithm resulting from injecting noise to each iteration of GD is usually called PGD (Perturbed gradient descent) that essentially minimizes an averaged objective function

$$R(\mathbf{x}) := \mathbb{E}_{\eta \sim N(0,\sigma^2 \mathbf{I})} E(\mathbf{x} + \eta).$$

For small values of $\sigma$, we can expand $R(\mathbf{x})$ into

$$R(\mathbf{x}) = E(\mathbf{x}) + \mathbb{E}_\eta \eta^T \nabla E(\mathbf{x}) + \frac{1}{2}\mathbb{E}_\eta \eta^T \nabla^2 E(\mathbf{x})\eta + O(\sigma^3) = E(\mathbf{x}) + \frac{1}{2}\sigma^2 Tr(\nabla^2 E(\mathbf{x})) + O(\sigma^3),$$

where $Tr$ denotes the trace operator. Thus minimizing $R(\mathbf{x})$ regularizes the trace Hessian of $E$. When minimizing $R(\mathbf{x})$ using an SGD type of update, the iterations would be

$$\mathbf{x}^{k+1} = \mathbf{x}^k - h\nabla E(\mathbf{x}^k + \eta_k), \quad \text{where} \quad \eta_k \sim \mathcal{N}(\mathbf{0}, \sigma^2 \mathbf{I}),$$

which is known as a form of PGD. Because of the finite learning rate, the updates would follow the modified flow with an IGR term, which in this case is

$$E(\mathbf{x} + \eta_k) + \frac{h}{4}\|\nabla E(\mathbf{x} + \eta_k)\|^2.$$

In expectation, the modified loss is

$$\mathbb{E}_{\eta_k}\left[ E(\mathbf{x} + \eta_k) + \frac{h}{4}\|\nabla E(\mathbf{x} + \eta_k)\|^2) \right]$$

$$= \mathbb{E}_{\eta_k}\left[ E(\mathbf{x}) + \eta_k^T \nabla E(\mathbf{x}) + \frac{1}{2}(\eta_k)^T \nabla^2 E(\mathbf{x})\eta_k \right]$$

$$+ \frac{h}{4}\mathbb{E}_{\eta_k}\|\nabla E(\mathbf{x}) + \nabla^2 E(\mathbf{x})\eta_k + \nabla^3 E(\mathbf{x})[\eta_k, \eta_k]\|^2) + O(\sigma^3)$$

$$= E(\mathbf{x}) + \sigma^2 Tr(\nabla^2 E(\mathbf{x})) + \frac{h}{4}\left[ \sigma^2 \|\nabla^2 E(\mathbf{x})\|_F^2 + \mathbb{E}_{\eta_k}\|\nabla E(\mathbf{x}) + \nabla^3 E(\mathbf{x})[\eta_k, \eta_k]\|^2 \right] + O(\sigma^3).$$

We see that now there is a Hessian regularization term $\frac{h}{4}\sigma^2 \|\nabla^2 E(\mathbf{x})\|_F^2$ coming out of IGR which does not vanish even around local minimizers, and it's strength is proportional to the learning rate. We expect this new regularization term to get stronger when momentum is added, as momentum amplifies the power of IGR (Remark 4.2, 5.3). This observation suggests that IGR and noise injection as two different types of implicit regularization might be able to reinforce each other when used collaboratively.

## 8 CONCLUSION

This work studies the generalization of momentum driven gradient descent approach through the lens of implicit regularization (IR) with both theoretical analysis and experimental validation provided. We examined the similarities and differences between (SGD) and (SGD+M) and find that (SGD+M) with suitable parameters outperforms (SGD) in almost all settings. Moreover, we found that in addition to momentum, IGR may also be magnified by noise injection, which is a topic we want to further explore in the future.

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
