# OpenReview forum: "Implicit regularization in Heavy-ball momentum accelerated stochastic gradient descent"
_ICLR.cc/2023/Conference — ICLR 2023 notable top 25%_

### Official Review · Reviewer_Yonc · 2022-10-21

**Confidence:** 4
**Correctness:** 3
**Technical Novelty And Significance:** 1
**Empirical Novelty And Significance:** 2
**Recommendation:** 6

**Clarity, Quality, Novelty And Reproducibility:**

This paper is generally speaking written with good clarity.

Regarding originality, I’d like to hear from the authors about the above before making a final assessment.

In terms of quality, I also have an additional question, which I did not count as weakness: on page iii, is it really necessary to involve PAC-Bayes? Can one directly say something about the landscape? Note that Hessian (and thus flatness) has already been quantitatively related to hyperparamters in, e.g., [Bock & Weiß. Local Convergence of Adaptive Gradient Descent Optimizers. 2021] and [Wang et al. Large Learning Rate Tames Homogeneity. 2021], and while I understand the setups may be slightly different, I wonder if the same technique of local stability analysis cannot be applied.

**Strength And Weaknesses:**

Strength:

(1) the problem considered is of critical importance to machine learning.

(2) working out the algebra to quantify the role of momentum variable is nice.

(2) the empirical discussions and experiments are interesting.

Weakness:

(1) Modified equation for SGD with momentum, unlike suggested, has already been studied in details. See for instance [Li et al. Stochastic modified equations and adaptive stochastic gradient algorithms. 2017] for a short version and [Li et al. Stochastic modified equations and dynamics of stochastic gradient algorithms i: Mathematical foundations. 2019] for a long version where the validity of modified equation, which is otherwise just a formal result, is actually rigorously quantified.

(2) In addition, unlike the aforementioned results, the derivation of modified equation in this paper is only formal. In fact, the formal asymptotic expansion used for deriving modified equation will blow up once h becomes not that small. This problem is already starting to be addressed by the more recent literature, such as [Wang et al. Large Learning Rate Tames Homogeneity. 2021].

(3) The main results, Theorem 4.1  and 5.1, are just short time results, meaning $T$ is fixed in Theorem 4.1 and $n \leq T/h$ for fixed $T$ in Theorem 5.1. The derivation is based on standard (but still nice) numerical analysis, which in fact requires $T$ to be O(1) (i.e. small) in order for the hidden constants in the error bounds to be not exponentially large. This limits the applicability of the results to practical training, unless the setup is more specific so that early stopping becomes reasonable or necessary.

In terms of actions, it would be great if the above are at least fairly discussed in the paper.


**Summary Of The Paper:**

This paper uses a classical tool in numerical analysis, namely modified equation, to account for the effect of small but finite learning rate used in (stochastic) gradient descent with momentum. As a result of this analysis and as a standard treatment, the continuous time limit of the optimizer, which is an ODE, will have additional O(h) term(s), which can then be employed to characterize the optimizer’s implicit bias. This paper investigates in such correction term(s) and shows that the momentum parameter also affects the strength of the implicit regularization. Empirical discussions and experiments are also provided.

**Summary Of The Review:**

This paper is considering an interesting problem from a nice quantitative perspective. However, before I can better understand the innovation, I’m afraid I cannot give a very positive recommendation.

---

> ### Author Response · Authors · 2022-11-07
> **Response to Reviewer Yonc [Part-1 out of 3]**
>
> We thank the reviewer for the constructive comments and for taking the time to thoroughly read this paper. We are glad that you find our work to be of critical importance to the community and you think that the paper is written with good clarity. Below, we address your comments on weakness point-by-point and highlight the originality of our work which seems to be your prime concern.
>
> **Comment**:      *Modified equation for SGD with momentum, unlike suggested, has already been studied in details. See for instance [1] for a short version and [2] for a long version where the validity of modified equation, which is otherwise just a formal result, is actually rigorously quantified.*
>
>
> **Response**:  Thanking you for giving us the chance to clarify our contribution.
>
> - Compared with [1,2], the main difference is that they proved **weak-sense approximation** to the Ito SDE ($\alpha$-weak-sense approximation of $x$ by $X$ means $\max_n |\mathbb{E}(x_{n})-\mathbb{E}(X(n\Delta t))| <  C(\Delta t)^{\alpha}$) while our result is in **strong sense approximation** ($\max_n |(x_{n}-X(n\Delta t)| <  C(\Delta t)^{\alpha}$). Weak-sense approximation ensures the closeness *over the distribution of the sample processes* $x$ and $X$ whereas strong-sense approximation requires the *actual sample-paths of the two processes* to be close, per realization of the random process. We refer Definition-2 of [2] for a more rigorous definition of weak sense approximation. So, the IGR term we found is effective in every  training instance and with any fixed initialization of weights, while the IGR in their paper is effective only in distribution,  meaning that one needs to train the network many times with different initialization and random draws of the batches, and then take the average. Considering in deep learning, we usually only train the model once,  the strong-sense IGR approximation we obtain better explains what happens in practice.  It informs us that our derived IGR term exists in and therefore guides each instance of training, instead of only on average.
> -  Our result provided an explicit mathematical formula for IGR for each training instance of SGD+M, and we explained how it is linked to generalization through the PAC argument. [1,2] didn't derive explicit IGR formula for SGD+M, although they did derive it for SGD. Both [2] and [3] derived IGR for SGD, but the closeness is in the weak sense ( in expectation or in distribution). We derived the IGR formula for both SGD and SGD+M, and the closeness to the modified loss (with IGR) is measured in the strong sense. In fact to the best of our knowledge, no other work (before us) derived the strong sense approximation even for just SGD.
> -   In fact, we deem both [2] and [3,4] as prior work to ours, and cited both of them in the manuscript. But we feel our general motivation is closer to [3,4], as it also emphasizes the relationship between the IGR term and the generalization. But we agree that [2] derived similar forms of IGR for SGD and also appeared earlier (2019), so we would like to add more detailed discussion and comparison with it in the paper.
> -   This might be a minor issue compared to the points above: The result in [2] requires extra conditions on the initial step (Theorem 3 condition 1 in the longer version). Since we really need order-2 approximation to preserve the IGR information, the $\alpha$ in Theorem 3 of the above paper has to be 2, and the $\rho(\epsilon)$ has to be $\eta^2$, which then makes it not very obvious whether condition (1) of Theorem 3 still possibly hold.
> -   To summarize, the main technical contribution of this current paper is the following. We observed that the reason why the technique in [5] cannot give us an $O(h^2)$ closeness for SGD+M is that it tries to find IGR through linking it to a second-order ODE, but the ODE is actually only $O(h)$-close to the discrete H.B momentum update. We then observe that the technique in [2] cannot give a strong sense of approximation, as it looks for IGR by linking it to Ito SDE, the latter uses the Wiener process, so it cannot be pointwise close to the momentum updates. Therefore, we propose to study the IGR through the piece-wise first-order ODE, **which is both $O(h^2)$ close and close in the strong sense approximation to discrete SGD+M updates**.

---

> > ### Author Response · Authors · 2022-11-07
> > **Response to Reviewer Yonc [Part-2 out of 3]**
> >
> > **Comment** :   *In addition, unlike the aforementioned results, the derivation of modified equation in this paper is only formal. In fact, the formal asymptotic expansion used for deriving modified equation will blow up once h becomes not that small. This problem is already starting to be addressed by the more recent literature, such as [6].*
> >
> > **Response**: Yes, our result needs $h$ to be small enough for the high order terms to be negligible. We do not think it is a proof artifact, instead, we think it is a cost we have to pay to derive the instance-wise IGR. Other papers [5] and [4] with the target of deriving the instance-wise IGR all need to have this assumption.
> >
> > Thanks for pointing us to this interesting paper, which shows that when GD is applied to the matrix factorization problem, we can use a learning rate of up to $\frac{4}{L}$, larger than the normal stability threshold $\frac{2}{L}$. We feel this result is very much like a special case of the edge-of-stability [7] phenomenon, which shows what happens when a larger than normal learning rate is used to a train general network. The observation there is that a larger than $\frac{2}{L}$ learning rate does not necessarily make the algorithm blow up, instead, the network will find a way to  decrease the sharpness ($L$) until the current (fixed) learning rate falls into the stable region (see Fig-1).  The exact reason for this phenomenon is not known yet, but there are some new insights recently [8]. Overall, when $h$ (in terms of $L$) is allowed to be large, it enters the ``edge of stability region" where the Taylor expansion approach (like ours) stops working. More complicated analysis is needed to get a result of this regime.
> >
> >
> > **Comment**: *The main results, Theorem 4.1 and 5.1, are just short time results, meaning $T$  is fixed in Theorem 4.1 and $n \leq \frac{T}{h}$  for fixed $T$ in Theorem 5.1. The derivation is based on standard (but still nice) numerical analysis, which in fact requires $T$ to be O(1) (i.e. small) in order for the hidden constants in the error bounds to be not exponentially large. This limits the applicability of the results to practical training, unless the setup is more specific so that early stopping becomes reasonable or necessary.*
> >
> >
> > **Response**:  Yes, since we used first-order ODEs to derive the IGR,  the exponential constant is unavoidable as it is a typical term that shows up when proving the existence of ODE solutions.  So, indeed we have to assume $T$ to be $O(1)$, a constant not necessarily small, but definitely cannot go to infinity.  We would like to mention that in previous works, $T$ is also not allowed to go to infinity, as it is contained in the constant of the upper bounds in both Theorem 3 of [2]  and Theorem 2 of [5]. And we'd like to mention that these two papers have other assumptions that we do not have.
> >
> >
> > The IGR term is much more significant in the early part of training and becomes weak near convergence (usually occurs at large T) to a local minima (as $||\nabla E(x)||_2^2 \rightarrow 0$). Hence we believe a large $T$ would not limit the effect IGR and IGR-M has on practical training. Even though theoretically, we cannot prove a small error in IGR for when T gets big, in practice,  big T usually means the algorithm is near convergence and thus the trajectories are converging (so the difference between them would probably not continue to increase). Also, as we discussed in section 7,  the IGR term in later iterations is very weak so is not that important, it is the IGR in early iterations that makes a difference in the final prediction error.
> >
> >
> > **Comment**: *Is it really necessary to involve PAC-Bayes? Can one directly say something about the landscape? Note that Hessian (and thus flatness) has already been quantitatively related to hyperparameters in, e.g [9] and [6], and while I understand the setups may be slightly different, I wonder if the same technique of local stability analysis cannot be applied.*
> >
> > **Response**:  Since PAC-Bayes bound is arguably the tightest upper bound we currently have on the generalization error, it is  the most common approach to study generalization. Small spectral norm of the Hessian, or flatness, is also commonly used to ensure good generalization, but rigorously speaking, it is a derived condition from the PAC-Bayes bound. For our problem,  due to the special form of the IGR, it is easier for us to draw a connection with the PAC-Bayes bound than with the norm of the Hessian.
> > To the best of our knowledge, the local stability analysis is used to analyze the stability and convergence of an algorithm instead of the generalization. Although we believe that there might be some relation between stability and generalization, it is not well-understood yet based on the current literature.

---

> > > ### Author Response · Authors · 2022-11-07
> > > **Response to Reviewer Yonc [Part-3 out of 3]**
> > >
> > >
> > > We have addressed all the points and tried our best to highlight the originality and the motivation of our work. Please let us know if you have any further questions regarding our work. We are happy to engage in further discussion.
> > >
> > >
> > >
> > > **References**
> > >
> > > -1) Li, Qianxiao, Cheng Tai, and E. Weinan. "Stochastic modified equations and adaptive stochastic gradient algorithms." International Conference on Machine Learning. PMLR, 2017.
> > >
> > > -2) Li, Qianxiao, Cheng Tai, and E. Weinan. "Stochastic modified equations and dynamics of stochastic gradient algorithms i: Mathematical foundations." The Journal of Machine Learning Research 20.1 (2019): 1474-1520.
> > >
> > > -3) Smith, Samuel L., et al. "On the origin of implicit regularization in stochastic gradient descent." arXiv preprint arXiv:2101.12176 (2021).
> > >
> > > -4) Barrett, David GT, and Benoit Dherin. "Implicit gradient regularization." arXiv preprint arXiv:2009.11162 (2020).
> > >
> > > -5) Kovachki, Nikola B., and Andrew M. Stuart. "Continuous time analysis of momentum methods." Journal of Machine Learning Research 22.17 (2021): 1-40.
> > >
> > > -6) Wang, Yuqing, et al. "Large learning rate tames homogeneity: Convergence and balancing effect." arXiv preprint arXiv:2110.03677 (2021).
> > >
> > > -7)  Cohen, Jeremy M., et al. "Gradient descent on neural networks typically occurs at the edge of stability." arXiv preprint arXiv:2103.00065 (2021).
> > >
> > > -8) Damian, Alex, Eshaan Nichani, and Jason D. Lee. "Self-Stabilization: The Implicit Bias of Gradient Descent at the Edge of Stability." arXiv preprint arXiv:2209.15594 (2022).
> > >
> > > -9) Bock, Sebastian, and Martin Georg Weiß. "Local Convergence of Adaptive Gradient Descent Optimizers." arXiv preprint arXiv:2102.09804 (2021).

---

> > > > ### Comment · Reviewer_Yonc · 2022-11-19
> > > > **Thank you for the clarifications**
> > > >
> > > > I've changed my recommendation from 3 to 6.

---

### Official Review · Reviewer_Cu2v · 2022-10-23

**Confidence:** 3
**Correctness:** 4
**Technical Novelty And Significance:** 3
**Empirical Novelty And Significance:** 3
**Recommendation:** 8

**Clarity, Quality, Novelty And Reproducibility:**

This work is generally well-written and has a nice flow of ideas.

Some typos:
- Mismatched parentheses: Theorem 4.1. ((IGR-M) ; Equation (11)
- Remark 5.4, adjusted learning rate -> effective learning rate
- Section 6, (GD+M) performed for various β with a fixed step-size h − 0.1, -> h = 0.1


**Strength And Weaknesses:**

This is a very interesting work, which extends the idea of implicit gradient regularization (IGR) (Barrett & Dherin, 2020) to explain the effect of momentum in GD/SGD. The theorems explain the empirical observation that with the same effective learning rate, the convergence of GD/SGD is more unstable (has larger variance) than GD+M/SGD+M. And also the experiments in (Ma, J., & Yarats, D., 2018) which fixed the effective learning rate, although they used Nesterov's momentum (maybe IGR can also be used to study Nesterov's momentum? or even QHM?). The authors give very detailed discussion and insights about the theorems, which I found very nice. I didn't check all the proofs, but they looks correct.

Section 7 points out another very interesting topic, though without empirical verifications. From my perspective, the current contribution is sufficient for acceptance. I think it would be interesting to also discuss the effect of learning rate schedulers. The authors mentioned that the IGR term will get very weak when converging, and thus the generalization performance of GD/SGD and GD+M/SGD+M might be similar in this case. This can be observed in (Ma, J., & Yarats, D., 2018) for NAG and SGD after learning rate reductions. I think that the IGR is a powerful tool, and that the authors are moving in the correct direction using it to understand NN optimizers.



(Ma, J., & Yarats, D., 2018)  Ma, J., & Yarats, D. (2018). Quasi-hyperbolic momentum and Adam for deep learning. In International Conference on Learning Representations.

**Summary Of The Paper:**

This work studies the generalization performance of momentum methods (GD+M, SGD+M) through the lens of implicit gradient regularization (Barrett & Dherin, 2020). The authors show that the effect of momentum can be understood as stronger implicit gradient regularization + stronger variance reduction (compared with GD/SGD using the same effective learning rate), and verify their claim with numerical justifications. These results provide a deeper understanding of the mechanism of momentum and some promising future directions (such as combining implicit gradient regularization with noise injection).

**Summary Of The Review:**

This work explains the effect of momentum using the idea of implicit gradient regularization (IGR). The theorems nicely explain several empirical differences between GD/SGD and GD+M/SGD+M. I believe that the IGR has the potential to explain more NN optimizers, and thus the authors are heading in the correct direction. I vote for acceptance.

---

> ### Author Response · Authors · 2022-11-07
> **Response to Reviewer Cu2v**
>
> We thank the reviewer for the valuable comments. We are glad that you like our work and like us, feel that IGR has the potential to explain deep neural network optimizers. We also thank you for pointing out the typos and we will be careful to correct them and proof-read a few more times to get rid of other typos.
>
> **Comment** *maybe IGR can also be used to study Nesterov's momentum? or even QHM?*
>
> **Response** Yes, we think so too. We include this as a future work and believe it is indeed  possible to address how NNs perform with Nesterov's momentum.
>
> **Comment**  *I think it would be interesting to also discuss the effect of learning rate schedulers. The authors mentioned that the IGR term will get very weak when converging, and thus the generalization performance of GD/SGD and GD+M/SGD+M might be similar in this case. This can be observed in [1] for NAG and SGD after learning rate reductions.*
>
> **Response** We agree with your insightful observation and thank you for pointing us to this interesting work. During our experiments we indeed observed a very similar phenomenon as in Figure-1 and 2 of [1], which can be seen in one of our experiments (as in Figure-2 in our appendix). Kindly refer to section 5.2 in appendix for our recently added experiment on scheduler effects. The initial difference (before scheduler comes into effect) in test accuracy between (SGD) and (SGD+M) is well explained by IGR (Remark-5.3 in manuscript). But after scheduler decreases $h$, the effect of IGR in both (SGD) and (SGD+M) gets diminished (as $IGR \propto h $). However, from empirical observations (both in the Fig-2 of our appendix and Fig 1,2 of [1]) the difference in test accuracy of (SGD) and (SGD+M) (near convergence) still exists but may not be in a pronounced way as the initial iterations.
>
> We believe this is because during the earlier iterations, the significantly stronger IGR for (SGD+M) guides it's trajectory through flatter sub-manifolds than that of (SGD). The effect is  prominent enough that even after scheduler is activated (also near convergence), (SGD+M) still has a slightly higher test accuracy than (SGD).
>
> We are glad that you raised an interesting point. Learning rate scheduler is commonly used in practice and we plan to include a subsection in the appendix explaining this phenomenon. Also, please let us know if there are any other questions you would like us to address. We are happy to engage in further discussion. Thank you.
>
> **References**
>
> 1) Ma, Jerry, and Denis Yarats. "Quasi-hyperbolic momentum and Adam for deep learning." arXiv preprint arXiv:1810.06801 (2018).

---

### Official Review · Reviewer_5gUB · 2022-10-24

**Confidence:** 3
**Correctness:** 3
**Technical Novelty And Significance:** 3
**Empirical Novelty And Significance:** 2
**Recommendation:** 6

**Clarity, Quality, Novelty And Reproducibility:**

The paper is well-structured and well-motivated, and a pleasant to read. The high-level approach follows prior works' analysis but there are some novel idea to get around the difficulty in directly extending. Some part of the writing feels a little less rigorous and a better balance between explaining intuition and providing rigor derivations may be helpful.

**Strength And Weaknesses:**

The paper has a strong motivation, clean exposition and nice writing. I think it answers the very interesting problem of implicit regularization effect of momentum steps from one perspective, which may have positive effects on future work toward a more complete understanding of implicit regularization and acceleration. The following parts could be improved a bit more:

1) The theoretical part seems to suggest the parameter beta of heavy-ball momentum shall be set to as high as possible, since then it leads to both faster convergence and better generalization. This I feel may be counter-intuitive to observations in practice. While the authors did try to touch upon such concerns in last sentences of Section 4, it may be helpful to particularly touch upon that point and how the theory guide practical choice of beta in more detail.

2) The convergence rate difference of 1/(1-beta) factor seems to be somewhat different from the classic square root speedup for Polyak's momentum method in convex optimization. Is there a good intuition explaining why this is true?

3) The sharpness and "edge of stability" explanation on middle of page 7 seems a bit hand-wavy. It might be helpful to elaborate in a bit more detail why this means (GD+M) shall potentially have worse test accuracy.



**Summary Of The Paper:**

The paper studies the implicit regularization effect in GD and SGD methods with momentum. It quantifies the regularization effect when algorithms take heavy-ball momentum step with a parameter beta, which partially explains the generalization ability and why momentum helps GD and AGD to find flatter minima. The authors also corroborate their findings with some empirical results.

**Summary Of The Review:**

I like the problem it studies and the perspective it takes. With some efforts the exposition can be made better and I'd be happy to re-evaluate once my concerns have been properly addressed.

---

> ### Author Response · Authors · 2022-11-07
> **Response to Reviewer 5gUB (Part-1 out of 2 )**
>
> We thank the reviewer for providing some very useful suggestions on how to create a better balance between explaining intuition and providing rigor derivations. We have addressed each of your comments point-by-point and will update our manuscript based on your suggestion.
>
> **Comment** : *The theoretical part seems to suggest the parameter beta of heavy-ball momentum shall be set to as high as possible, since then it leads to both faster convergence and better generalization. This I feel may be counter-intuitive to observations in practice. While the authors did try to touch upon such concerns in last sentences of Section 4, it may be helpful to particularly touch upon that point and how the theory guide practical choice of beta in more detail.*
>
> **Response**: Our IGR result is obtained by truncating high order terms in h. In fact, the high order terms also contain $\frac{1}{(1-\beta)}$ (we didn't make it explicit as $\beta$ is assumed to be a constant throughout), so if $(1-\beta) \rightarrow 0$, the truncation is too large and our result would no longer hold. Therefore, after $\beta$ reaches some threshold, it is not necessarily the larger the better.  In this paper, we follow the tradition in [1]  to fix $\beta$ and let $h$ be the variable, so we denoted the high order terms as $O(h^m)$ and $\beta$ is absorbed in the big O notation. A future direction would be letting both $\beta$ and $h$ be variables, and examine their relations. That might give more insight of how to choose $\beta$.
>
>
> **Comment**: *The convergence rate difference of $\frac{1}{(1-\beta)}$ factor seems to be somewhat different from the classic square root speedup for Polyak's momentum method in convex optimization. Is there a good intuition explaining why this is true?*
>
> **Response**: Thanks for this very insightful question. We note that the convergence rate comparison can only be made when a given  effective learning rate makes both GD  and GD+M converge. In other words, the effective learning rate has to be less than $2/L$ where $L$ is the sharpness $|| \nabla^2 E ||$.
> As is well known (e.g., see this blog [2]), the square root speed up is achieved by setting $h = \frac{4}{(\sqrt{L} + \sqrt{\mu})^2}$, and $\beta = (\frac{\sqrt{\kappa} -1}{\sqrt{\kappa} +1})^2 $. Under this setting, the effective learning rate is $\frac{h}{(1-\beta)}  =\frac{1}{\sqrt{L\mu}}$, which is larger than $2/L$ provided the condition number $\kappa>4$ (highly likely the case in practice). So our result cannot provide a comparison between the two methods under this setting, as no choice of stable stepsize can make GD reach this effective learning rate achieved by GD+M. This is why our results suggested due to the greater stability, momentum can achieve a higher effective learning rate than GD, that partially explains why GD+M is better than GD in terms of generalization, as larger learning rate usually leads to better generalization (through IGR [4]).
> On the other hand, if we use a sufficiently small $h$ and $\beta$  then we can indeed show (by classical convergence analysis) that the convergence rate of momentum is about $1/(1-\beta)$ times faster than GD, thus matching our theorem. We have added the proof here (after two threads) (https://openreview.net/forum?id=ZzdBhtEH9yB&noteId=1TyZM_92VL) and will also add it in the appendix of the manuscript.

---

> > ### Author Response · Authors · 2022-11-07
> > **Response to Reviewer 5gUB (Part-2 out of 2 )**
> >
> >
> > **Comment**: *The sharpness and 'edge of stability' explanation on middle of page 7 seems a bit hand-wavy. It might be helpful to elaborate in a bit more detail why this means (GD+M) shall potentially have worse test accuracy*
> >
> > **Response**: Sure. The `edge of stability' (EOS) [3] is a  phenomenon that shows during network training by full batch gradient descent, the sharpness $|| \nabla^2 E ||_{2} $ tends to progressively increase until it reaches the threshold $\frac{2}{h}$ and then hovers around it. For GD+M, the sharpness can increase to a higher level of $\frac{2(1+\beta)}{h}$. Since larger sharpness usually means worse generalization, the EOS theory then potentially predicts that adding momentum hurts the generalization.
> >
> > However, limited understanding of EOS in current literature prevents us from commenting to which extent
> > EOS can predict generalization. In particular, EOS is not observed in training with SGD or SGD+M, and is not significant  for shallow networks or networks trained with cross-entropy loss. Moreover, the theoretical explanation of the driving force of progressive sharpening is
> > missing and the numerical experiments on GD+M is very limited.
> >
> > For GD+M, it has been mostly observed (in the ML community) that it has better generalization performance than (GD), which aligns with our numerical results and contradictory to the EOS prediction.
> > From this viewpoint we believe IGR-M is a stronger candidate in explaining the superior generalization performance of (GD+M) over (GD) as the (IGR-M) theory goes hand-in-hand with empirical observations.
> >
> > We would like to elaborate this in the manuscript.
> >
> >
> > Finally, we would like to thank you for raising some good points that will potentially improve the manuscript making it more reader-friendly. We are also glad that you think that our work is well motivated and may have positive effect on understanding role of implicit regularization in generalization. Please let us know if you have any further comments or questions regarding the paper. We are happy to engage in further discussion.
> >
> > **References**
> >
> > -1) Kovachki, Nikola B., and Andrew M. Stuart. "Continuous time analysis of momentum methods." Journal of Machine Learning Research 22.17 (2021): 1-40.
> >
> > -2) https://trungvietvu.github.io/notes/2018/Momentum
> >
> > -3) Cohen, Jeremy M., et al. "Gradient descent on neural networks typically occurs at the edge of stability." arXiv preprint arXiv:2103.00065 (2021).
> >
> > -4) Barrett, David GT, and Benoit Dherin. "Implicit gradient regularization." arXiv preprint arXiv:2009.11162 (2020).

---

> > > ### Author Response · Authors · 2022-11-10
> > > **Proof:  In stable regime, convergence-rate of (GD+M) is $\frac{1}{(1-\beta)}$ larger than (GD)  using classical convergence analysis [Part-1 out of 2]**
> > >
> > > ### **Proof:In stable regime, convergence-rate of (GD+M) is $\frac{1}{(1-\beta)}$ larger than (GD)  using classical convergence analysis.**
> > >
> > >
> > > Classical convergence of (GD) and (GD+M) is considered in a locally quadratic surface. On a standard quadratic, the minimization is
> > >  $ min_{x} f(x)= \frac{1}{2} x^{T} A x -b^T x +c $,  where $A$ is positive semi-definite matrix with eigen-values in $[\mu,L]$. A simple change of variable would mean doing a minimization of the form $\min_{x} \frac{1}{2} x^T \Sigma x $, where $\Sigma$ contains the eigenvalues of A on the diagonal. Hence $\nabla f(x) = \Sigma x$ and $\nabla^2 f(x) = \Sigma$. Furthermore, the condition number of the objective function is denoted as $\kappa = \frac{L}{\mu}$. Throughout the proof $|| .||$ means $l_{2}$ norm unless specified.
> > >
> > > For Heavy-Ball method, the iterates follow:
> > > \begin{align}
> > >     x^{k+1} =x^{k}-h\nabla f(x^{k}) +\beta (x^{k}- x^{k-1})
> > > \end{align}
> > >
> > > On a locally quadratic, the iterates roughly follow
> > > \begin{align}
> > >      x^{k+1} = x^{k}-h\Sigma x^{k}  +\beta (x^{k}- x^{k-1}) = ((1+\beta)I - h \Sigma)x^{k} -\beta x^{k-1}
> > > \end{align}
> > > With slight rearrangement, it could be written as :
> > >
> > > \begin{align}
> > >    \begin{bmatrix}
> > > x^{k+1}\\\\
> > > x^{k}
> > > \end{bmatrix} = \begin{bmatrix}
> > > (1+\beta)I -h\Sigma & -\beta I \\\\
> > > I & \mathbf{0}
> > > \end{bmatrix} \begin{bmatrix}
> > > x^{k}\\\\
> > > x^{k-1}
> > > \end{bmatrix}
> > > \end{align}
> > >
> > > Denoting $y^{k}=  \begin{bmatrix}
> > > x^{k+1}\\\\
> > > x^{k}
> > > \end{bmatrix}$ and $T= \begin{bmatrix}
> > > (1+\beta)I -h\Sigma & -\beta I \\\\
> > > I & \mathbf{0}
> > > \end{bmatrix} $, the norm of $||y^{k} ||_{2} $ is derived as follows:
> > >
> > >
> > > \begin{align}
> > >     || y^{k}||_{2} =  ||T y^{k-1}||_2 =   || T^{k} y^{0}||_2 \leq  || T^{k}||_2 ||y^{0}||_2 \leq (\rho(T))^k \kappa(V)|| y^{0} ||_2
> > > \end{align}
> > >
> > > where $\rho(T) $ is the spectral radius of $T$, $T$ has an eigen-decomposition $T=VDV^{-1} $ and $\kappa(V)$ is the condition-number of $V$.
> > >
> > > $T$ is permutation-similar to the block diagonal matrix $T = \begin{bmatrix}
> > > T_{1} & \mathbf{0} &.& . & \mathbf{0}\\\\
> > >  \mathbf{0} & T_{2} & .& . & \mathbf{0}\\\\
> > > . & . & . & .&  .\\\\
> > >  \mathbf{0} & \mathbf{0} &. &. & T_{n}\\\\
> > > \end{bmatrix} $, where
> > >  $T_{j} = \begin{bmatrix}
> > > 1+\beta-\alpha \lambda_{j} & -\beta\\\\
> > > 1 & 0\\\\
> > >  \end{bmatrix} $ is a $2\times2 $ matrix for $j=1,2..n $. Letting $r_{j}$ denote the eigen-values of $T_{j} $ which would satisfy $    r_{j}=
> > > \begin{cases}
> > >    \frac{1}{2}((1+\beta -\alpha \lambda_{j})\pm \sqrt{ (1+\beta-h\lambda_{j})^2 -4\beta}),& \text{if } (1+\beta-h\lambda_{j})^2 -4\beta=\Delta_{j} > 0 \\\\
> > >       \frac{1}{2}((1+\beta -\alpha \lambda_{j}) \pm i \sqrt{|\Delta_{j}|},              & \text{otherwise}
> > > \end{cases} $
> > >
> > > where $i=\sqrt{-1}$.
> > > The convergence factor $\rho(T)$ is determined by the eigenvalue with the largest magnitude among all matrices $T_{j}$, i.e, $ \rho(T) =\max_{j} |r_{j}|= \max |r_{1}|,|r_{n}|$.
> > >
> > > Now depending upon the 4 conditions $\Delta_{j}\leq 0 \equiv \beta \geq (1-\sqrt{h \lambda_{j}}) $,  $\Delta_{j}> 0 \equiv \beta \leq  (1-\sqrt{h \lambda_{j}}) $ , $|1-\sqrt{h\mu}| < |1-\sqrt{h L}|$ and $|1-\sqrt{h\mu}| >|1-\sqrt{h L}|$, we have four sub-cases to determine $\rho(T)$:
> > >
> > > -  1) If  $0< h \leq (\frac{2}{\sqrt{L}+\sqrt{\mu}})^2 $ and $\beta \geq  (1-\sqrt{h\mu})^2$
> > > -   2) If $0< h \leq (\frac{2}{\sqrt{L}+\sqrt{\mu}})^2 $ and $\beta <  (1-\sqrt{h\mu})^2$
> > > -  3) If $h >(\frac{2}{\sqrt{L}+\sqrt{\mu}})^2 $ and $\beta \geq (\sqrt{h L}-1)^2 $
> > > - 4)  If $h >(\frac{2}{\sqrt{L}+\sqrt{\mu}})^2 $ and $\beta < (\sqrt{h L}-1)^2 $
> > >
> > > A small $h$ and fixed $\beta$ satisfies condition 2 and the effective learning rate lies in the stability regime of GD. Under this particular condition (2), we have $\Delta_{1}>0 $, hence the spectral radius  $\rho(T)$ becomes (by taking the larger $|r_{j}|$) :
> > >
> > >
> > > \begin{align}
> > >    & \rho^{(GD+M)} = \frac{1}{2}(1+\beta -h \mu + \sqrt{(1+\beta-h\mu)^2 -4\beta}) \quad  \text{[considering the larger term]}\\\\
> > >    & = \frac{1}{2}(1+\beta -h \mu + \sqrt{(1-\beta)^2-2h\mu(1+\beta) +h^2\mu^2 }) \\\\
> > >    &  = \frac{1}{2}(1+\beta -h \mu + (1-\beta)(\underbrace{\sqrt{1-\frac{2h\mu(1+\beta) +h^2\mu^2}{(1-\beta)^2} }}_{1-\frac{1}{2}\frac{2h\mu(1+\beta)}{(1-\beta)^2} +O(h^2) }  -1 ) + (1-\beta))\\\\
> > >    & \approx \frac{1}{2} (1+\beta -h \mu -\frac{h\mu(1+\beta)}{(1-\beta)} +  (1-\beta) ) \quad  \text{[small $h$ approximation]}\\\\
> > >    & = 1- \frac{h \mu}{(1-\beta)}
> > > \end{align}
> > >
> > > Similarly, for (GD) with learning-rate $\tilde{h}$ minimizing a locally quadratic function, using the classical convergence approach, we have $||x^{k}|| \leq (\rho^{GD})^{k} ||x^{0}|| $   , where $ \rho^{GD} =\max (|1-\tilde{h}\mu|,|1-\tilde{h}L| )$.
> > >
> > > Hence for a small enough $\tilde{h}$ i.e,( $0< \tilde{h} \leq \frac{2}{L+\mu} $), we have for the convergence rate for GD to be :
> > > \begin{align}
> > >     & \rho^{GD} = 1-\tilde{h}\mu
> > > \end{align}
> > >
> > > [continued in the next thread]

---

> > > > ### Author Response · Authors · 2022-11-10
> > > > **In stable regime, convergence-rate of (GD+M) is $\frac{1}{(1-\beta)}$ larger than (GD)  using classical convergence analysis [Part-2 out of 2]**
> > > >
> > > > Putting $\tilde{h} = \frac{h}{(1-\beta)}$, we see that $\rho^{(GD+M)} \approx \rho^{(GD)} $. Which means if we use a learning rate $\frac{1}{(1-\beta)}$ times larger for GD, it will match the convergence rate of (GD+M).
> > > >
> > > > Equivalently under the same learning rate for (GD) and (GD+M) (say $h$), the convergence rate of (GD+M) is $\frac{1}{(1-\beta)}$ times larger than that of (GD),i.e, $\rho^{(GD+M)} \approx \frac{1}{(1-\beta)} \rho^{(GD)} $.
> > > >
> > > > The proof follows basic convergence analysis of GD and GD+M on quadratic functions and derivations here closely follows from  https://trungvietvu.github.io/notes/2018/Momentum.

---

> ### Author Response · Authors · 2022-11-23
> **Modifications to manuscript**
>
> Dear reviewer,
>
> Thank you for your insightful comments that helped us improve our paper. Since, we first posted response to your comments, we made a couple of modifications and fixed some typos in the proof. We have included these modifications in our mansucript and updated the paper accordingly (before the deadline 18th Nov).
>
>   We would be grateful if you could let us know if the modifications made in the paper addresses your questions. Thank you!

---

### Official Review · Reviewer_DtPB · 2022-10-29

**Confidence:** 4
**Correctness:** 4
**Technical Novelty And Significance:** 4
**Empirical Novelty And Significance:** 2
**Recommendation:** 8

**Clarity, Quality, Novelty And Reproducibility:**

- The paper is largely clearly written, minus the early focus on O(h^2) that I have talked about in weaknesses.

- The result and the analysis is novel and original as far as I know. However, I have not gone through all the proofs.

- The authors have provided the code for their experiments, and from the presented experiments, I dont see an obvious problem with reproducibility.

**Strength And Weaknesses:**

Strengths:

- The paper explains the setup well, explains the implicit regularization effect due to the closeness to the modified flow, and also why the modified flow is suited for better generalization which helps in a lot in getting a complete picture of the impact of this work.
- The variance reduction effect of momentum in SGD is one of the most interesting aspects of the presented analysis. The authors have made relevant comments how the overall generalization is likely due to a combination of various factors including variance, stability and the contributing implicit regularization. The effect of variance, for example, still requires further imho even though there are a couple of studies cited in this work that talk about how smaller variance can lead to bad local minima.
- The relevant literature is well-cited and well discussed, mentioning key contributions and how they relate to this paper.
- The empirical section is largely sufficient and complements the theoretical contributions well.

Weaknesses:
- I had a hard time reading the paper the first time mostly because I had to wait till eq 6 on page 4 before finally understanding and appreciating what O(h^2) closeness means and why that is important, even though the authors use it freely starting from right in the abstract. I suggest the authors consider re-wording this. I have not read all the related works cited in this paper, so I am not sure if it is a common jargon used in this community, but nevertheless it makes the paper a lot harder to read for more general audience.
- Is there a reason why the authors used only the simple linear 2layer model to illustrate the IGR effect in Sec 4.1, especially considering that they have mentioned earlier in the paper about jelassi and Li 2022 not applicable to more general NNs? I understand they reproduced the setting of Barrett and Dherin 2020 but it would have been interesting to see a non-linear activation. Nevertheless, this is not a deal-breaker since the stability and full-batch experiments complement the theoretical contributions well.




**Summary Of The Paper:**

This paper provides a new analysis for implicit regularization for the heavy ball momentum gradient descent. The paper shows that the momentum updates for gradient descent and its stochastic counterpart are closer to the modified regularized gradient flow than previously known. Their analysis is based on studying a piecewise first order ODE, as opposed to a second order ODE analysis that was undertaken by a previous work which makes the crucial difference in the final results.

**Summary Of The Review:**

The paper makes a non-trivial contribution to the field and should be of interest to the community. The result demonstrating the regularizing effect of heavy ball in both the classical gradient descent and the stochastic settings is interesting, and so is the presented analysis.

---

> ### Author Response · Authors · 2022-11-07
> **Response to Reviewer DtPB (Part-1 out of 2 )**
>
> We thank the reviewer for the constructive comments. We are glad that you believe that this work can bring a valuable contribution to the field especially in understanding generalization abilities of momentum methods. We have carefully gone through your comments and addressed each of them point-by-point. We believe addressing these points in the manuscript would indeed make the paper more reader-friendly.
>
> **Comment**: *The variance reduction effect of momentum in SGD is one of the most interesting aspects of the presented analysis. The authors have made relevant comments how the overall generalization is likely due to a combination of various factors including variance, stability and the contributing implicit regularization. The effect of variance, for example, still requires further imho even though there are a couple of studies cited in this work that talk about how smaller variance can lead to bad local minima.*
>
> **Response**: Thank you for the comment. As rightly pointed out, in remark-5.6 we cite out some important works that state that a larger variance in mini-batch gradients is important for optimizer to reach good local minimas. Therefore, we think the variance reduction effect of SGD+M indeed does not help with generalization in this case. But stronger IGR for (SGD+M) helps in achieving better generalization than (SGD) and can hence explain why we observe better generalization for (SGD+M).
>
> For the last sentence in your comment, would you like us to include a more intuitive explanation as to why smaller variance may lead to bad local minimas? Did you actually mean 'the effect of variance, for example, still requires
> further *explanation* imho even though ..'? (quoting your comment).
> If so, we can include a brief explanation as follows.
>
> -  Deep neural network training is highly non-convex containing a lot of local minimas and valleys. A good optimizer should have the property of escaping multiple bad minimizers to settle for a good local minima (preferrably flatter and having a lower loss value).
> -  In SGD, the mini-batch gradient can be thought of as noise in  + actual gradient at that point (full-batch gradient) $\nabla E_{i}(x) = \nabla E(x)  +\eta$. So, when an optimizer is stuck in a valley having a bad local minima, the noisy gradient $\nabla E_{i}(x)$ does not point to the direction of the local minima of that valley. Rather the noise $\eta$ can provide it a direction to *escape* the valley (having a bad local minima).
> -   Very recently, this intution has been mathematically formalized by [1]. Here in theorem-2, authors show that Escape efficiency of SGD  $ \propto \exp({-\frac{B}{h}\Delta E \lambda_{max}^{-\frac{1}{2}}}) $  where $B$, $h$, $\Delta E$ and $\lambda_{max}$ denote the batch-size, learning rate, depth of minima and largest eigen value of Hessian  respectively. In short, a smaller batch-size (B) and a larger learning rate are crucial to escape bad local minimas.
>
> We will be happy to address and emphasize the role of variance in training deep networks in our appendix (or manuscript is space permits).
>
> **Comment**: *I had a hard time reading the paper the first time mostly because I had to wait till eq 6 on page 4 before finally understanding and appreciating what $O(h^2)$ closeness means and why that is important, even though the authors use it freely starting from right in the abstract. I suggest the authors consider re-wording this. I have not read all the related works cited in this paper, so I am not sure if it is a common jargon used in this community, but nevertheless it makes the paper a lot harder to read for more general audience.*
>
> **Response**: Again, thank you for this valuable comment. We understand your concern and we indeed want the readers to understand the importance of $O(h^2)$-closeness which we believe is the essence to understand the work. We will be careful to reword our manuscript and explain it more carefully at the start.

---

> > ### Author Response · Authors · 2022-11-07
> > **Response to Reviwer DtPB (Part-2 out of 2)**
> >
> >
> > **Comment**: *Is there a reason why the authors used only the simple linear 2-layer model to illustrate the IGR effect in Sec 4.1, especially considering that they have mentioned earlier in the paper about jelassi and Li 2022 not applicable to more general NNs? I understand they reproduced the setting of Barrett and Dherin 2020 but it would have been interesting to see a non-linear activation.*
> >
> > **Response**: We wanted to provide an example that is simple enough to clearly demonstrate   the advantage of GD+M over GD with the effective learning rate. We chose the 2D example as it can be easily plotted. We could have added ReLU but for this example, we noted that adding a ReLU does not make a difference, since everything is positive.
> >
> >
> >
> > Lastly, we thank the reviewer for his/her valuable feedback which will help a lot to make our work more reader-friendly.
> >
> > **References**
> >
> > - [1]Ibayashi, Hikaru, and Masaaki Imaizumi. "Quasi-potential theory for escape problem: Quantitative sharpness effect on SGD's escape from local minima." arXiv preprint arXiv:2111.04004 (2021).

---

> > > ### Author Response · Authors · 2022-11-23
> > > **Manuscript updated**
> > >
> > > Dear reviewer,
> > >
> > > We thank you for your review that helped us improve our manuscript. Based on your suggestions, we made the following modifications before the deadline (18th Nov):
> > > -  We added Section-7 in Appendix (https://tinyurl.com/bdzeeeww) explaining how a larger variance improves generalization.
> > > - We added a formal definition on  closeness in the strong sense (Definition-3.1) in manuscript.
> > > - We extended the linear 2D experiment to a non-linear one with a Sigmoid layer (https://tinyurl.com/2p9s38sc). This can be found in section-8 of the appendix. We have also submitted the jupyter notebook for reproducing our result.
> > >
> > > Please let us know if the modifications addressed your questions and comments. Thank you !

---

> > ### Comment · Reviewer_DtPB · 2022-11-16
> > **Further details on Variance vs generalization**
> >
> > Yes, please. IMHO adding further discussion on the role of variance should be useful to the reader.

---

### Author Response · Authors · 2022-11-18
**Summary of modifications and changes**

Dear chairs and reviewers,

We sincerely thank you for taking the time to thoroughly read our paper and providing important feedback that helped improve the quality of the paper. We would also like to thank all the reviewers for acknowledging the importance of this work to the ML community. Here, we summarize and highlight the changes we incorporated in the manuscript based on each reviewer's suggestion. **The changes in the manuscript are highlighted in red**.

**1) Formal definition of $O(h^{\alpha})$ closeness in the strong sense**

The concept of $O(h^2)$ closeness in the strong sense is a key concept in our paper. We apologize that the lack of this definition in the first draft hindered Reviewer **DtPB** from understanding the paper when reading the first time. Based on both reviewer **Yonc** and **DtPB**'s suggestion, we added a formal  definition on $O(h^{\alpha})$ closeness in the *strong sense* (Definition-3.1) in manuscript. We hope this will help the readers:
- appreciate the importance of $O(h^2)$ closeness of the modified equation.
- highlight the novelty of our work and distinguish our work from [1] which proved weak-sense approximation.

**2) Adding a convergence comparison between (GD) and (GD+M) for the quadratic problem**

Reviewer-**5gUB** provided a valuable insight that the *'convergence rate difference of the $\frac{1}{(1-\beta)}$ factor seems to be somewhat different from the classic square root speedup for Polyak's momentum method in convex optimization.'* Here, we explained why our result is consistent with prior knowledge of the convergence rate of the two methods, which is well known to the optimization community. Our result implies that the momentum is $\frac{1}{(1-\beta)}$ faster than GD. In the proof, we show that this matches the convergence rates obtained from the classical convergence analysis of the two methods in the regime of parameters considered in this paper. The proof can be found in Section-6 of Appendix (https://tinyurl.com/mpen2zdk).

**3) Additional experiments on learning-rate schedulers**

Based on Reviewer-**Cu2v** 's valuable suggestion, we added an experiment that shows the effect of IGR in SGD and (SGD+M) when a learning-rate scheduler is applied. The empirical observations here are consistent with our theory (IGR-M) as discussed in Section-5.2 of Appendix (https://tinyurl.com/582vmpw2).

**4) Adding a discussion of the influence of variance reduction on generalization**

In the appendix, we added Section-7 (https://tinyurl.com/bdzeeeww) explaining how a larger variance improves generalization based on reviewer-**DtPB** 's suggestion.

**5) Adding an additional 2D-experiment with non-linear activation**
As suggested by Review **DTPB**, we extended the linear 2D experiment to a non-linear one with a Sigmoid layer. As can be found in Section-8 of the appendix (https://tinyurl.com/2p9s38sc), the result for the nonlinear 2D example again confirms our theory that the IGR in (GD+M) is stronger than that in GD. We also provided the code of this experiment in a Jupyter notebook file for reproducibility.

**6) Discussing future directions**

As suggested by Reviewer **Yonc**, we added Section-9 in the appendix (https://tinyurl.com/2eptkbpv) to address the limitation of our IGR-based analysis. We mentioned that our current analysis is based on a fixed $\beta$ value and in practice a bound asymptotic in both $h$ and $\beta$ ($h \rightarrow 0$, $\beta \rightarrow 1$) would be a future work. This also addresses Reviewer-**5gUB**'s concern that increasing $\beta$ does not necessarily improve generalization beyond a certain threshold.

**References**
1) Li, Qianxiao, Cheng Tai, and E. Weinan. ''Stochastic modified equations and dynamics of stochastic gradient algorithms i: Mathematical foundations'' The Journal of Machine Learning Research 20.1 (2019): 1474-1520.

Lastly, it goes without saying that we, the authors are more than happy to engage in further discussion. We would again like to thank all the reviewers and the area-chair for their time and effort in evaluating this work.

---

### Decision · Program_Chairs · 2023-01-20

**Decision:**

Accept: notable-top-25%

**Justification For Why Not Higher Score:**

This is a nice paper, but the bar for oral is very high.

**Justification For Why Not Lower Score:**

Calibrated according to my other meta-reviews, this paper has an interesting perspective on a problem that has not been considered before (afaik), and the reviewer reception overall is positive.

**Metareview: Summary, Strengths And Weaknesses:**

This paper studies the implicit bias of SGD with a specific type of momentum, showing that as the step size is taken to zero, the updates are close to an ODE on a modified loss function.  This is an interesting perspective on momentum, and combined with the positive reception by the reviewers, I feel this is a clear accept.

**Note From Pc:**

if the above contains the word "oral" or "spotlight" please see: "oral" presentation means -> notable-top-5% and "spotlight" means -> notable-top-25%. As stated in our emails, we are disassociating presentation type from AC recommendations